# Low-noise GaAs quantum dots for quantum photonics

Liang Zhai [1✉], Matthias C. Löbl [1], Giang N. Nguyen[1,2], Julian Ritzmann [2], Alisa Javadi [1], Clemens Spinnler [1], Andreas D. Wieck [2], Arne Ludwig [2] & Richard J. Warburton [1]

Quantum dots are both excellent single-photon sources and hosts for single spins. This combination enables the deterministic generation of Raman-photons—bandwidth-matched to an atomic quantum-memory—and the generation of photon cluster states, a resource in quantum communication and measurement-based quantum computing. GaAs quantum dots in AlGaAs can be matched in frequency to a rubidium-based photon memory, and have potentially improved electron spin coherence compared to the widely used InGaAs quantum dots. However, their charge stability and optical linewidths are typically much worse than for their InGaAs counterparts. Here, we embed GaAs quantum dots into an *n-i-p*-diode specially designed for low-temperature operation. We demonstrate ultra-low noise behaviour: charge control via Coulomb blockade, close-to lifetime-limited linewidths, and no blinking. We observe high-fidelity optical electron-spin initialisation and long electron-spin lifetimes for these quantum dots. Our work establishes a materials platform for low-noise quantum photonics close to the red part of the spectrum.

[1] Department of Physics, University of Basel, Klingelbergstrasse 82, CH-4056 Basel, Switzerland. [2] Lehrstuhl für Angewandte Festkörperphysik, Ruhr-Universität Bochum, DE-44780 Bochum, Germany. ✉email: liang.zhai@unibas.ch

Quantum dots (QDs) in III–V semiconductors form excellent sources of indistinguishable single-photons. These emitters have a combination of metrics (brightness, purity, coherence, repetition rate) which no other source can match[1–4]. These excellent photonic properties can be extended by trapping a single electron to the QD, enabling spin-photon entanglement[5] and high-rate remote spin-spin entanglement creation[6]. Underpinning these developments are, first, a self-assembly process to create nano-scale QDs; and second, a smart heterostructure design along with high-quality material. The established platform consists of InGaAs QDs embedded in GaAs. However, the InGaAs QDs emit at wavelengths between 900 and 1200 nm, a spectral regime lying inconveniently between the telecom wavelengths (1300 nm and 1550 nm) and the wavelength where silicon detectors have a high efficiency[7] (600–800 nm). It is important in the development of QD quantum photonics to extend the wavelength range towards both, shorter and longer wavelengths.

GaAs QDs in an AlGaAs matrix can be self-assembled by local droplet etching[8,9] and have a spectrally narrow ensemble[10,11]. They emit at wavelengths between 700 and 800 nm. This is an important band: it coincides with the peak quantum efficiency of silicon detectors; it contains the rubidium $D_1$ and $D_2$ wavelengths (795 nm and 780 nm, respectively) offering a powerful route to combining QD photons with a rubidium-based quantum memory[12]. Furthermore, GaAs QDs have typically more symmetric shapes, facilitating the creation of polarisation-entangled photon pairs from the biexciton cascade[4,13].

GaAs QDs have also very low levels of strain[9,14–17]. In contrast, the high level of strain in InGaAs QDs complicates the interaction of an electron spin with the nuclear spins on account of the atomic site-specific quadrupolar interaction[14,18]. For electro-statically defined GaAs QDs, the spin-dephasing time, $T_2^*$, has been prolonged to the micro-second regime by narrowing the nuclear spin distribution together with real-time Hamiltonian estimation[19]. Applied to a droplet GaAs QD, such techniques could prolong the spin dephasing time to values several orders of magnitude above the radiative lifetime. In this case, in combination with optical cavities[20], droplet GaAs QDs can potentially serve as fast, high-fidelity sources of spin-photon pairs and cluster states[21].

The development of GaAs QDs for quantum photonics lags far behind the InGaAs QDs. Recurrent problems are blinking[22,23] (telegraph noise in the emission) and optical linewidths well above the transform limit[13,16,23–25]. Both of these problems are caused by charge noise. On short time-scales, the charge environment is static such that successively emitted photons exhibit a high degree of coherence[4,25]. On longer time-scales, however, the charge noise introduces via blinking an unacceptable stochastic character to the photon stream. An additional weak non-resonant laser provides control over the noise to a certain extend, though it does not remove the blinking completely[22].

For InGaAs QDs, embedding the QDs in an $n$-$i$-$p$ diode has profound advantages: the charge state is locked by Coulomb blockade[26–28]; the charge noise is reduced significantly[29]; and the exact transition frequency can be tuned in-situ via a gate voltage[3,30]. Such a structure is missing for GaAs QDs[13,16,22–25]—in previous attempts, charge-stability was not demonstrated[31,32]. A materials issue must be addressed: the barrier material AlGaAs must be doped, yet silicon-doped AlGaAs contains DX-centres[33,34] which both reduce the electron concentration, causing the material to freeze out at low temperatures, and lead to complicated behaviour under illumination. Here, we resolve this issue—all doped AlGaAs layers have a low Al-concentration. In this case, the DX level lies above the conduction band minimum and thus is unoccupied at cryogenic temperatures[33]. The QDs are grown in a region with higher Al-concentration, which is well-established for the growth of these QDs[8]. On GaAs QDs in this device we demonstrate charge-control via Coulomb blockade, optical linewidths just marginally above the transform limit, blinking-free single-photon emission, electron spin initialisation, and a spin-relaxation time as large as ~50 µs.

## Results

**Sample design and characterisation**. The sample is grown on a GaAs-substrate with (001)-orientation. Below the active region of the sample, a distributed Bragg reflector is grown to enhance the collection efficiency of the photons emitted by the QDs. The QDs are embedded in an $n$-$i$-$p$-diode structure where the QDs are tunnel-coupled to the $n$-type layer. The $n$-type back gate consists of silicon-doped $Al_{0.15}Ga_{0.85}As$. The low Al-concentration in this layer is crucial to avoid the occupation of DX-centres in $n$-type AlGaAs[33,34]. A tunnel barrier consisting of 20 nm $Al_{0.15}Ga_{0.85}As$ followed by 10 nm $Al_{0.33}Ga_{0.67}As$ separates the QDs from the $n$-type back gate. The QDs are grown in the $Al_{0.33}Ga_{0.67}As$-layer by using local droplet-etching[8]. The QD-density is $n_{QD} = 0.37 \pm 0.01$ µm$^{-2}$. Above the QDs, there is 274 nm of $Al_{0.33}Ga_{0.67}As$ followed by a $p$-type top gate. The top gate is composed of carbon-doped $Al_{0.15}Ga_{0.85}As$, where reduced Al-concentration is used as well. A schematic bandstructure of the diode is shown in Fig. 1a; all Al-concentrations in this design are small enough that processing into micropillars[35] and nanostructures will not be hindered by oxidation[36]. In Table 1, the design of the full heterostructure is given.

We characterise our device by measuring the photoluminescence from a single QD as a function of the gate voltage, $V_g$, applied to the diode (Fig. 1b). As a function of $V_g$, the emission lines show a pronounced Stark-shift. At specific gate voltages, discrete jumps in the emission spectrum take place: one emission line abruptly becomes weaker and another line appears. This effect is the characteristic signature of charge-control of a QD via Coulomb blockade[26]: the net-charge of the QD increases one by one and the emission energy is shifted due to the additional Coulomb interaction with the new carrier.

We fit the relation $E = E_0 + \alpha F + \beta F^2$ to the dependence of the emission energy, $E$, on electric field, $F$ (Supplementary Fig. 2). The energy jumps between different charge plateaus are removed for the fit. We find $\alpha/e = 0.21$ nm, the permanent dipole moment in the growth direction, and $\beta = -1.35 \times 10^{-6}$ eV(kV/cm)$^{-2}$, the polarisability of the QD[37]. Extrapolating the fit shows that the Stark shift is zero at a non-zero electric field ($F = 7.8$ kVcm$^{-1}$). The non-zero value of $\alpha$ represents a small displacement between the "centre-of-mass" of the electron and the hole wavefunctions. The hole wavefunction is slightly closer to the back gate than the electron wavefunction.

**Resonance fluorescence from GaAs QDs**. We identify the neutral exciton, $X^0$, from its characteristic fine-structure splitting, as well as a quantum-beat in time-resolved resonance fluorescence (Supplementary Fig. 3). For our device, the fine-structure splittings are distributed over a range of 1–3 GHz (see Supplementary Fig. 4c). The fine-structure splittings are comparable to literature values on (001)-oriented samples[4,8]. Smaller fine-structure splittings can be obtained by using (111)-oriented samples[13] and strain-tuning[38]. We identify the other charge-states by counting the number of jumps in the emission spectrum as the gate-voltage increases/decreases. We measure emission from highly charged excitons ranging from the two-times positively charged exciton, $X^{2+}$, to the eight-times negatively charged exciton, $X^{8-}$. Such a wide range of charge tuning was not previously achieved with any QDs emitting in the close-to-visible wavelengths. Our GaAs QDs

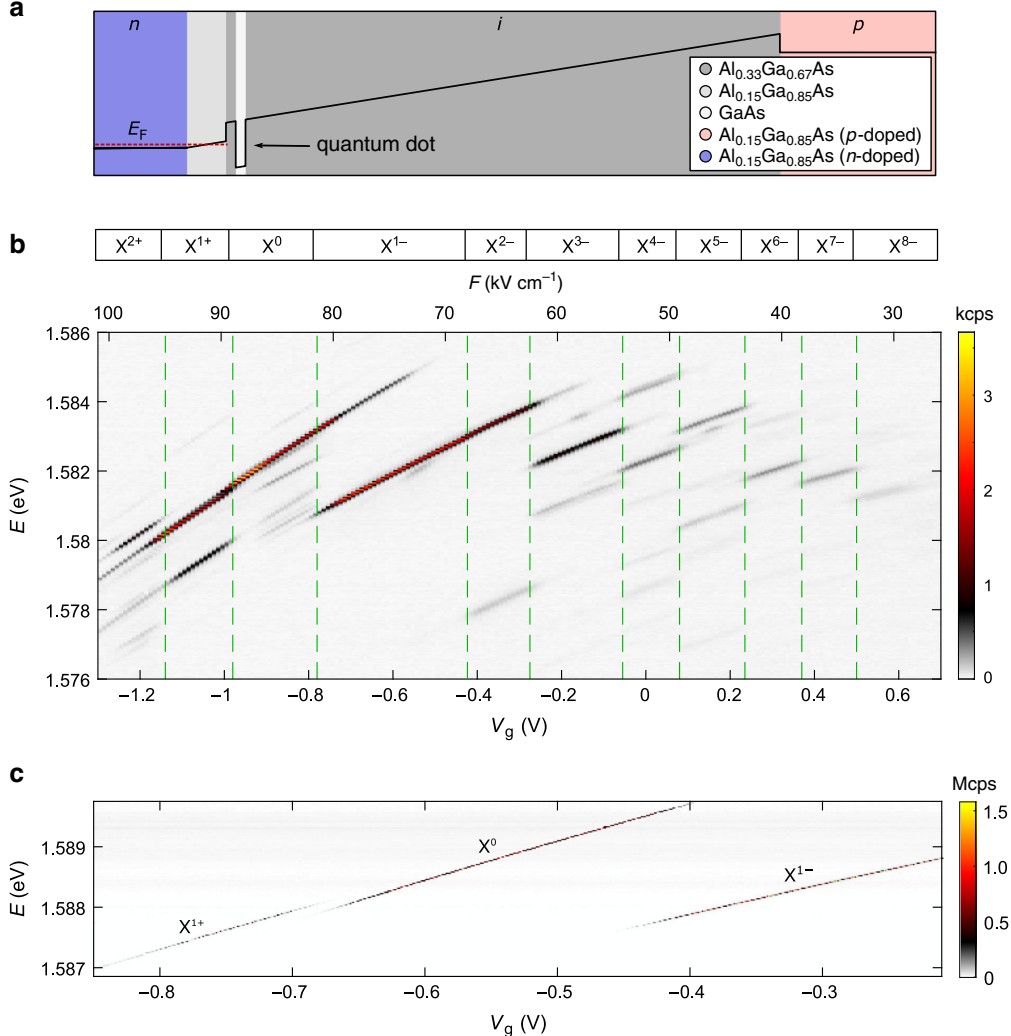

**Fig. 1 Tuning the charge state of single GaAs quantum dots. a** Schematic band structure (conduction band) of the diode hosting charge-tunable GaAs quantum dots. **b** The photoluminescence emitted by an exemplary single quantum dot as a function of the gate voltage, $V_g$. (Positive gate voltage indicates a forward bias.) The corresponding electric field, $F$, is plotted as an additional x-axis on top. The photoluminescence is resolved in energy by a spectrometer and measured on a CCD-camera. The emission spectrum shows several plateaus corresponding to different charge states of the quantum dot. We observe narrow photoluminescence-linewidths on highly charged excitons where up to eight additional electrons occupy the quantum dot. **c** Resonance fluorescence from $X^{1+}$, $X^0$, and $X^{1-}$ charge plateaus measured on another quantum dot (QD1). $X^{1+}$, $X^0$, and $X^{1-}$ represent the positive trion, the neutral exciton, and the negative trion, respectively. The measurement is performed by sweeping the gate voltage for different laser frequencies. The resonance fluorescence intensity is measured with a superconducting nanowire single-photon detector. This measurement is performed by resonant continuous-wave excitation below saturation. In saturation, the maximum count rate is 6.5 MHz (see Supplementary Fig. 1 for the power saturation curve).

give a large range of charge tuning due to their relatively large size[8] in comparison to the widely used InGaAs QDs[39].

We turn to resonant excitation. This excitation scheme is key for creating low-noise photons and represents a true test of the fidelity of the device as, unlike photoluminescence, continuum states are not deliberately occupied. By sweeping both the gate voltage and excitation laser frequency, we map out three charge plateaus of a single quantum dot (QD1) – $X^{1+}$, $X^0$, and $X^{1-}$ (see Supplementary Fig. 5 for photoluminescence of QD1). As is visible in Fig. 1c, the exact transition energy of all three charge states can be tuned via $V_g$ across a range of above 1 meV. At a fixed gate voltage, we determine a resonance fluorescence linewidth of $X^{1-}$ to be $0.64 \pm 0.01$ GHz (full width at half maximum) on scanning a narrow-bandwidth laser over the trion resonance (see Fig. 2a). (resonance fluorescence laser scans on $X^{1+}$ and $X^0$ are shown in Supplementary Fig. 3). This measurement takes several minutes: the linewidth probes the

sum of all noise sources over an enormous frequency bandwidth[40]. The measured linewidth is very close to the lifetime-limit of $\Gamma_r/2\pi = 0.59 \pm 0.01$ GHz. (It is assumed here the decay is radiative. The radiative decay rate $\Gamma_r$ is determined by recording a decay curve following pulsed resonant excitation, Fig. 2b). This result shows that there is extremely little linewidth broadening due to noise in our device. These excellent results are not limited to one individual QD. Shown in Fig. 2d is a linewidth measurement on a second QD (QD2). In the central part of the $X^{1-}$ charge-plateau (from $V_g = -0.5$ V to $V_g = -0.4$ V in Fig. 2c), we also measure a close-to lifetime-limited linewidth. On average, the ratio between the measured linewidth and the lifetime limit is 1.08 for QD2. At the edges of the charge-plateau, the linewidth increases—a well-know effect due to a co-tunnelling interaction with the Fermi-reservoir[41]. Comparably good properties are found for in total seven out of ten randomly chosen QDs with $X^{1-}$ below 785 nm (see Supplementary Fig. 4a, b).

**Table 1 Sample design with relevant growth parameters.**

| Material | Thickness (nm) | Temperature (°C) | Duration (s) | Comments |
|---|---|---|---|---|
| GaAs:C | 5 | 540 | 25.1 | p++-doped epitaxial gate |
| $Al_{0.15}Ga_{0.85}As$:C | 10 | 540 | 42.7 | p++-doped epitaxial gate |
| $Al_{0.15}Ga_{0.85}As$:C | 65 | 540 | 277.7 | p+-doped epitaxial gate |
| $Al_{0.33}Ga_{0.67}As$ | 273.6 | 540 | 921.8 | Blocking barrier |
| GaAs | 2 | 605 | 10 | Filling of the etched nano-holes |
| – | – | 605 | 60 | Droplet etching |
| Al | – | 605 | 3.7 | Al-droplet 0.9 nm plus 1 ML Al[a] |
| $Al_{0.33}Ga_{0.67}As$ | 10 | 590 | 33.7 | Tunnel barrier (high Al) |
| $Al_{0.15}Ga_{0.85}As$ | 15 | 590 | 64.1 | Tunnel barrier (low Al) |
| $Al_{0.15}Ga_{0.85}As$ | 5 | 575 | 21.4 | Tunnel barrier (low-temperature) |
| $Al_{0.15}Ga_{0.85}As$:Si | 150 | 590 | 640.8 | n+-doped back gate[b] |
| $Al_{0.15}Ga_{0.85}As$ | 50 | 590 | 209.3 | Buffer layer |
| AlAs/$Al_{0.33}Ga_{0.67}As$ | 10× (67.08/59.54) | 590 | 8904.7 | Distributed Bragg reflector |
| GalAs/AlAs | 22× (2.8/2.8) | 590 | 1101.7 | Short-period superlattice |
| GaAs | 100 | 590 | 601.8 | Start |

[a]For the Al-layer, the amount of deposited aluminium is given as the thickness of a corresponding AlAs-layer. The aluminium is deposited in an arsenic-depleted ambience.
[b]In the molecular beam epitaxy chamber used here, the background impurity concentration is estimated to be ~ $5 \times 10^{14}$ cm$^{-3}$ for $Al_{0.33}Ga_{0.67}As$ layers[50]. The doping concentration is around $2 \times 10^{18}$ cm$^{-3}$ for the n+ layer, while for p+ and p++ layers, it is around $2 \times 10^{18}$ cm$^{-3}$ and $8 \times 10^{18}$ cm$^{-3}$, respectively. Between the n-type back gate and the p-type top gate, the sample has a built-in potential of 1.82 V.

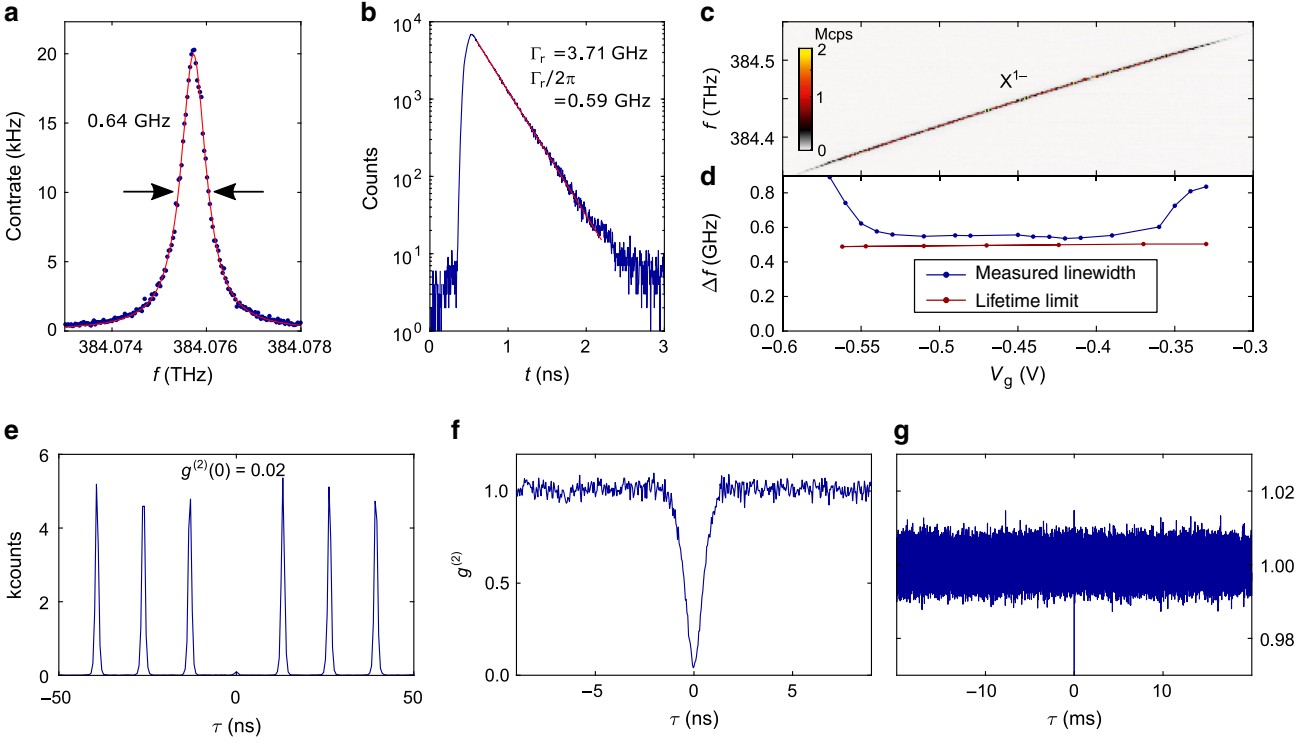

**Fig. 2 Time-resolved lifetime and photon-correlation measurements. a** Resonance fluorescence linewidth measured on the singly-charged exciton, $X^{1-}$ (QD1). The measurement is performed by sweeping a narrow-bandwidth laser over the $X^{1-}$ resonance. The overall time for the shown scan is ~8 min. A Lorentzian function (red line) fits perfectly to the data (blue dots), showing an optical linewidth of 0.64 ± 0.01 GHz. **b** Lifetime measurement on $X^{1-}$ under pulsed resonant excitation. The gate voltage is the same as in **a**. The measured decay rate ($\Gamma_r = 3.71 \pm 0.04$ GHz, corresponding to a lifetime of $1/\Gamma_r = 270 \pm 3$ ps) implies a lifetime-limited linewidth of $\Gamma_r/2\pi = 0.59 \pm 0.01$ GHz (Exponential fit). **c** Resonance fluorescence of $X^{1-}$ (QD2) as a function of the gate voltage. **d** Resonance fluorescence linewidth along with the lifetime-limit (obtained from separate lifetime measurements at the corresponding gate voltages). Similar to QD1, the linewidth of QD2 stays very close to the lifetime limit in the plateau centre. **e** Auto-correlation ($g^{(2)}$) measured under resonant $\pi$-pulse excitation. **f** Auto-correlation of the resonance fluorescence measured under weak continuous-wave excitation shown on a short time-scale. The $g^{(2)}$-measurement is normalised[44] by dividing the number of coincidences by its expectation value $T \cdot t_{bin} \cdot x_1 \cdot x_2$, where $T$ is the overall integration time, $t_{bin}$ is the binning time, and $x_1$, $x_2$ are the count-rates on the two single-photon detectors. **g** The same auto-correlation measurement as in **f** but evaluated on a much longer time-scale (milliseconds). The perfectly flat $g^{(2)}$ reveals the absence of blinking.

A remarkable feature is that the close-to-transform limited linewidths are observed despite the large dc Stark shifts of these QDs. Within the $X^{1-}$ plateau of QD1 (Fig. 1c), the dc Stark shift is 0.0347 GHz per V cm$^{-1}$, about a factor of four larger than the typical dc Stark shifts of InGaAs QDs[40]. The sensitivity of the transition frequency to the electric field renders the QD linewidth susceptible to charge noise. The close-to-transform limited linewidths reflect therefore an extremely low level of charge

noise in the device. Assuming that the slight increase in broadening with respect to the transform limit arises solely from charge noise, the linewidth measurement places an upper bound of ~3.0 Vcm$^{-1}$ for the root-mean-square (rms) electric field noise at the location of QD1. This upper bound is comparable to the best gated InGaAs QD devices[20,29,40,42,43].

For applications as single-photon source, it is crucial to demonstrate that the photons are emitted one by one, i.e., photon anti-bunching. Therefore, we continue our analysis by performing an intensity auto-correlation of the resonance fluorescence. This $g^{(2)}$-measurement is shown in Fig. 2e and Supplementary Fig. 6c, d for resonant $\pi$-pulse excitation with 76 MHz repetition rate. We observe a strong anti-bunching at zero time delay ($g^{(2)}(0) = 0.019 \pm 0.008$), corresponding to a single-photon purity of $1 - g^{(2)}(0)$ ~98%. The corresponding measurement under weak continuous-wave excitation is shown in Fig. 2f. ($g^{(2)}$-measurements versus excitation power, as well as laser detuning are mapped out in Supplementary Fig. 7, where clear Rabi oscillations are shown. In both cases, we find excellent agreement between the measured $g^{(2)}$ and a calculation based on a two-level model.) Also here, we observe a strong anti-bunching proving the single-photon nature of the emission.

Previous resonance fluorescence on GaAs QDs has suffered from blinking, i.e., telegraph noise in the emission[22]. This is a deleterious consequence of charge noise: either the QD charges abruptly or the charge state of a nearby trap changes, detuning the QD from the excitation laser in both cases. Blinking gives rise to a characteristic bunching ($g^{(2)} > 1$) in the auto-correction even for driving powers well below saturation[22]. We investigate this point here. Even out to long (millisecond) time-scales, the $g^{(2)}$-measurement is absolutely flat and close to one (see Fig. 2g). (We note that our analysis includes a mathematically justified normalisation of the $g^{(2)}$-measurement[44]). This result demonstrates that blinking is absent. This is a consequence both of the diode-structure, in particular Coulomb blockade which locks the QD charge, and the low charge noise in the material surrounding the QD.

We subsequently carried out $g^{(2)}$-measurements with either a small magnetic field along the growth direction or a laser slightly detuned from the QD resonance. In the former case the sensitivity to spin noise is enhanced, while in the latter case the sensitivity to charge noise is enhanced[40]. In Supplementary Fig. 8, we compare the $g^{(2)}$-measurements on millisecond time-scales. For the measurement with an additional magnetic field (Supplementary Fig. 8c, d), the $g^{(2)}$ remains flat and stays close to one. In contrast, we observe a small blinking when the laser is detuned (Supplementary Fig. 8e, f). We infer from these results that in our device charge noise is most likely to be responsible for the residual linewidth broadening.

**High-fidelity spin initialisation**. The diode structure allows us to load a QD with a single electron. The spin of the electron is a valuable quantum resource. To probe the electron-spin dynamics, we probe the X$^{1-}$ resonance fluorescence in a magnetic field (Faraday-geometry). In this configuration, the ground state is split by the electron Zeeman energy, and the excited state is split by the hole Zeeman energy (see Fig. 3a). As the diagonal transitions in this level-scheme are close to forbidden, the X$^{1-}$-charge-plateau splits into two lines which are separated by the sum of electron and hole Zeeman energies (see Fig. 3b). We find that the X$^{1-}$ charge-plateau becomes optically dim in its centre. This is the characteristic feature of spin-initialisation via optical pumping[27,43,45,46]. On driving e.g., the $|\uparrow\rangle - |\uparrow\downarrow,\Uparrow\rangle$ transition, the trion will most likely decay back to the $|\uparrow\rangle$-state via the dipole-allowed vertical transition. However, due to the heavy-hole

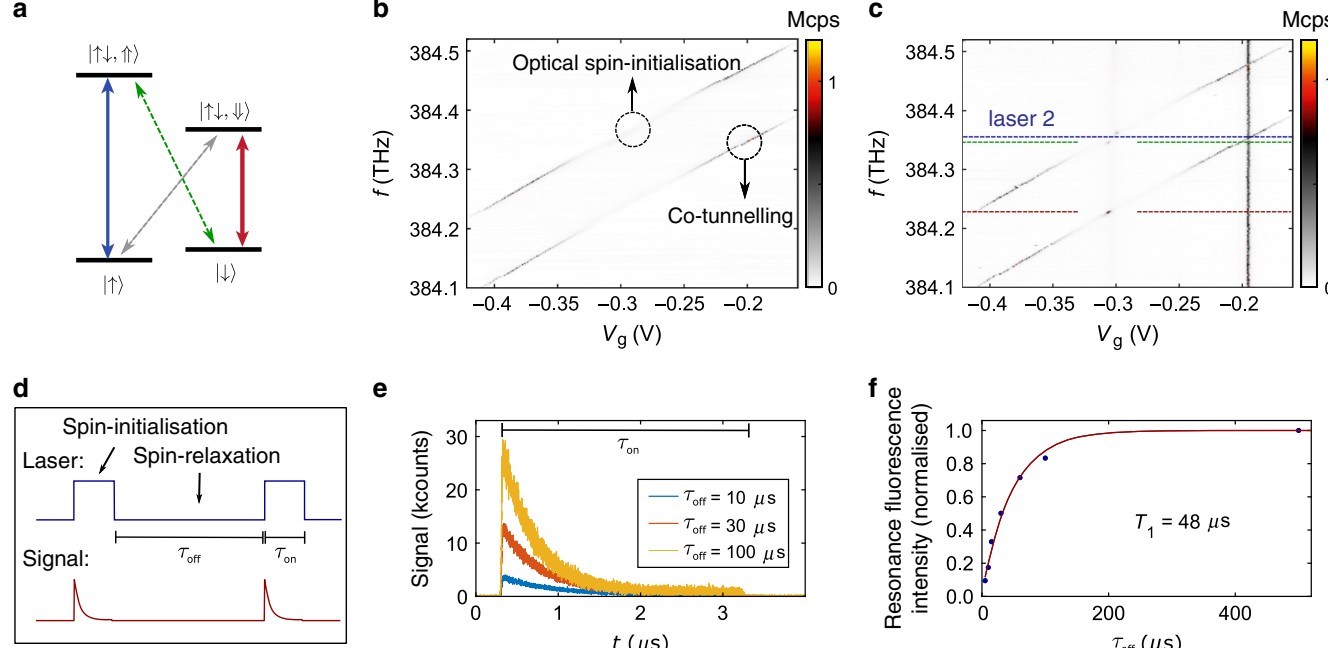

**Fig. 3 Initialisation of a single electron spin. a** Level scheme of the negative trion X$^{1-}$ in a magnetic field (Faraday geometry). **b** Optical spin-initialisation via optical pumping on X$^{1-}$. The measurement is carried out at $B = 6.6$ T. In the plateau centre, the resonance fluorescence disappears due to successful spin-initialisation; at the plateau edges it remains bright due to rapid spin-randomisation via co-tunnelling[41]. **c** Optical spin-initialisation and re-pumping with a second laser at a fixed frequency (laser 2). Recoveries of the signal are found in the plateau centre. **d** Schematic of the time-resolved spin-pumping measurement. **e** Resonance fluorescence intensity as a function of time. The signal drops due to optical spin-initialisation after turning the driving laser on. The overall intensity is larger when the time-delay $\tau_{off}$ between the laser pulses is larger. In this case, the electron spin has more time to relax back from the off-resonant state. **f** Resonance fluorescence intensity as a function of the waiting time between the spin-pumping laser pulses. The magenta line is an exponential fit to the data (blue dots). From this measurement we extract an electron-spin lifetime of $T_1$ ~48 ± 5 μs.

light-hole mixing or a weak in-plane nuclear field, it can also decay to the $|\downarrow\rangle$-state through the "forbidden" transition with a small probability. When the QD is in the $|\downarrow\rangle$-state, the driving laser is off-resonance on account of the electron Zeeman energy. Therefore, the centre of the $X^{1-}$-charge-plateau becomes dark and the initialisation of the electron spin in the $|\downarrow\rangle$-state is heralded by the disappearing resonance fluorescence. At the plateau-edges, resonance fluorescence reappears due to fast spin-randomisation via co-tunnelling[41]. By comparing the remaining intensity in the charge-plateau centre to the plateau edges[27], we estimate the spin initialisation fidelity to be $F = 98.3 \pm 0.3\%$. To confirm that the signal disappears in the plateau-centre on account of optical spin initialisation and not some other process, we perform a measurement with a second laser at a fixed frequency. When the fixed laser is resonant with $|\uparrow\rangle - |\uparrow\downarrow\Uparrow\rangle$ transition, we observe a recovery of the signal (Fig. 3c) on either driving the weak diagonal transition $|\downarrow\rangle - |\uparrow\downarrow\Uparrow\rangle$ or the strong vertical transitions $|\downarrow\rangle - |\uparrow\downarrow\Downarrow\rangle$ with the scan laser. While the fixed laser is tuned to $|\downarrow\rangle - |\uparrow\downarrow\Uparrow\rangle$ transition (at a different $V_g$), another recovery spot is seen as the scan laser drives the vertical transition $|\uparrow\rangle - |\uparrow\downarrow\Uparrow\rangle$. This confirms the optical spin-initialisation mechanism[27,45]. From the energy splitting at the plateau edges, we determine the electron and hole g-factors[17], $g_e = -0.076 \pm 0.001$ and $g_h = 1.309 \pm 0.001$. For the positively charged trion ($X^{1+}$), we also observe high-fidelity optical spin-initialisation (Supplementary Fig. 9) and narrow linewidths (0.62 GHz, see Supplementary Fig. 3), in this case of a hole spin.

How long-lived is the prepared spin state? To answer this question, we measure the time-dependence of the $X^{1-}$ spin initialisation[43,46]. The scheme is illustrated in Fig. 3d. First, we drive the $|\uparrow\rangle - |\uparrow\downarrow\Uparrow\rangle$ transition for $\tau_{on} = 3$ μs. During this laser pulse, the signal decreases due to optical spin-initialisation (Fig. 3e). Subsequently, we turn the laser off for a time $\tau_{off}$, and then turn the laser back on again. During the off-time the electron spin randomises. Fig. 3e shows that the resonance fluorescence signal is stronger when the waiting time $\tau_{off}$ is longer. The reason for this effect is that with increasing $\tau_{off}$ the spin has more time to randomise. For a short value of $\tau_{off}$, in contrast, the spin remains in the off-resonant state—it has no time to relax before the next optical pulse is applied. By measuring the signal strength for varying $\tau_{off}$ (Fig. 3f), we determine an electron-spin relaxation time of $T_1 = 48 \pm 5$ μs. Our result shows that the design of the tunnel-barrier between QDs and back gate is well suited for spin-experiments on single QDs. This $T_1$ value is significantly larger compared to the GaAs QDs without the n-i-p-diode structure[23]. The point is that the $T_1$ time is potentially longer than the coherence time $T_2$, such that the relaxation process governing $T_1$ is unlikely to limit the coherence time $T_2$[47].

## Discussion

In summary, we have developed charge-tunable GaAs QDs with ultra-low charge noise. We show notable improvements of the GaAs QDs properties: optical linewidths are close-to-lifetime-limited, blinking is eliminated, and long electron-spin lifetimes are achieved. From a materials perspective, the crucial advance is the new diode structure hosting GaAs QDs—a key feature is that all the doping is incorporated in layers of low Al-concentration. In this way, the occupation of DX-centres is avoided and the AlGaAs layers are conducting at low temperatures. The concepts developed in this work can be transferred to thinner diode-structures that allow integration into photonic-crystals and other nanophotonic devices[3,36]. From a quantum photonics perspective, our results pave the way to bright sources of low-noise single photons close to the red part of the visible spectrum. This will facilitate the developments of both short-range networks and a hybrid QD-rubidium quantum memory. On account of the low-strain environment in GaAs QDs, our work can also open the door to prolonged electron spin coherence.

## Methods

**Sample fabrication.** The sample heterostructure and the quantum dots are grown by molecular beam epitaxy (MBE). The MBE setup is similar to the one described in ref. [48]. The complete heterostructure of the sample is shown in Table 1. All doped layers in AlGaAs have low Al-concentration (<20%). The quantum dots are surrounded by AlGaAs with higher Al-concentration (33%), to enable the growth of QDs close to rubidium-frequencies and with small fine-structure splittings[8,11]. We fabricate separate Ohmic contacts to the n+ and p++ layers. For the n-type back gate, the sample is locally etched down by ∼360 nm in a mixture of sulfuric acid and hydrogen peroxide (concentrated $H_2SO_4$: 30% $H_2O_2$: $H_2O = 1$: 1: 50). NiAuGe is then deposited by electron-beam evaporation (with three steps: 60 nm AuGe (mass ratio 88:12), 10 nm Ni, and 60 nm AuGe), followed by thermal annealing at 370 °C for 60 s and 420 °C for 30 s. For the p-type top gate, a thin contact pad consisting of Ti (3 nm)/Au (7 nm) is evaporated locally on the top surface of the sample. Both contacts are electrically connected with silver paint.

**Experimental setups.** The sample is cooled down to 4.2 K in a liquid helium cryostat. We perform photoluminescence with a 632.8 nm He–Ne laser. The photoluminescence is collected by an aspheric objective lens (numerical aperture NA = 0.71) and sent to a spectrometer. Resonance fluorescence is performed with a narrow-band laser (1 MHz linewidth), using a cross-polarisation confocal dark-field microscope[22,49] to distinguish QD-signal from the scattered laser light. It is detected using superconducting-nanowire single-photon detectors and a counting hardware with a total timing jitter of ∼35 ps (full width at half maximum).

**Statistics of QD linewidths.** In our device, GaAs QDs with a small height (emission wavelength below ∼785 nm) tend to have excellent optical properties. We find that more than every second QD has a close to lifetime-limited linewidth (see Supplementary Fig. 4a,b). This includes QDs close to the [87]Rb $D_2$ line (∼780 nm). For QDs larger in size (emission wavelength above ∼785 nm), the QD linewidths are usually broader. The reason is probably the following: the GaAs QDs in our sample are grown by infilling nano-holes droplet-etched into a 10 nm-thin layer of $Al_{0.33}Ga_{0.67}As$ (see Table 1). The depths of the nano-holes, and therefore the heights of the QDs, typically range from 5 nm to 10 nm[8,11]. A QD emitting at higher wavelength tends to have a larger height[11]. When the height of a QD comes close to 10 nm, the optical properties could be affected by the $Al_{0.33}Ga_{0.67}As$/$Al_{0.15}Ga_{0.85}As$ interface. A simple solution is to make the $Al_{0.33}Ga_{0.67}As$-layer 5 nm thicker. In this case, we expect good optical properties also for QDs of higher wavelengths.

**Auto-correlation under different excitation schemes.** We investigate the stability of the QD under different excitation schemes. We start with continuous-wave (CW) excitation. We perform auto-correlation measurements on $X^{1-}$ at a constant gate voltage while exciting the QD with (i) an above-band laser ($\lambda = 632.8$ nm), (ii) a laser resonant with the p-shell, and (iii) a laser resonant with the s-to-s transition. The results are shown in (i) Supplementary Fig. 6a, (ii) Supplementary Fig. 6b, and (iii) Fig. 2g, respectively. In all three cases, the $g^{(2)}$ stays very flat and close to one—there is no blinking even on a long time-scale. This shows that the QD is a very stable quantum emitter under all three CW excitation schemes. From an applications point of view, it is usually necessary to drive the QD with a resonant pulsed laser. We investigate the auto-correlation under resonant π-pulse excitation in Fig. 2e. An evaluation of this $g^{(2)}$-measurement on a longer time-scale is plotted in Supplementary Fig. 6c, where the y-axis is displayed on a logarithmic scale to resolve the central peak. To investigate whether a strong π-pulse introduces any blinking, we plot the $g^{(2)}$-measurement in a histogram plot (Supplementary Fig. 6d) by summing up the coincidence events for every single pulse. This sum is divided by the expectation value for a perfectly stable source: the normalisation factor is $x_1 x_2 T_{int}/f_{rep}$, where $f_{rep}$ is the repetition rate of the pulsed laser, $x_1$, $x_2$ represent the count rates of the two detectors used for a $T_{int}$-long $g^{(2)}$-measurement. A derivation of the normalisation factor is given in Supplementary Fig. 6. Importantly, the histogram bars at non-zero time delay are flat and very close to one; the bar at zero delay is close to zero. This shows that the QD is a stable single-photon emitter for resonant π-pulse excitation.

**Potential noise source affecting the QD-linewidth.** The $g^{(2)}$-measurement shown in Fig. 2f,g is performed on a trion at zero magnetic field when the CW laser drives the QD resonantly. The sensitivity can be enhanced towards either spin noise or charge noise by applying a small magnetic field along the growth direction, and detuning the laser slightly from the QD-resonance by δ, respectively. A trion state is degenerate at zero magnetic field, consisting of two opposite spin ground states. When applying a magnetic field B, the degeneracy is lifted and the trion state is split into two by a Zeeman energy $E_z = g\mu_B B$, with g being the electron or hole g-factor, and $\mu_B$ the Bohr magneton. We maximise the spin noise sensitivity by applying a small magnetic field such that $E_z = \frac{\tilde{\Gamma}}{\sqrt{3}}$ (Supplementary Fig. 8c). Here $\tilde{\Gamma}$

represents the full width at half maximum (FWHM) of the QD emission. For the maximised spin noise sensitivity, the $g^{(2)}$-measurement does not show any clear sign of bunching (Supplementary Fig. 8d). The charge noise sensitivity is maximised when the laser is detuned from the QD by $\delta = \frac{\bar{\Gamma}}{2\sqrt{3}}$ (Supplementary Fig. 8e). In this configuration, we observe a small bunching peak in the $g^{(2)}$-measurement (Supplementary Fig. 8f). This result suggests that charge noise on a millisecond time-scale is responsible for the slight linewidth broadening.

## Data availability
The data that supports this work is available from the corresponding author upon reasonable request.

## Code availability
The code that has been used for this work is available from the corresponding author upon reasonable request.

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

## Acknowledgements
The authors thank Jan-Philipp Jahn and Armando Rastelli for stimulating discussions. L.Z. received funding from the European Union's Horizon 2020 Research and Innovation programme under the Marie Skłodowska-Curie grant agreement No. 721394 (4PHOTON). M.C.L., C.S., and R.J.W. acknowledge financial support from NCCR QSIT and from SNF Project No. 200020_156637. J.R., A.L., and A.D.W. gratefully acknowledge financial support from the grants DFH/UFA CDFA05-06, DFG TRR160, DFG project 383065199, and BMBF Q.Link.X 16KIS0867. A.J. acknowledge support from the European Union's Horizon 2020 research and innovation programme under the Marie Skłodowska-Curie grant agreement No. 840453 (HiFig).

## Author contributions

L.Z., M.C.L., G.N.N., A.J., and C.S. carried out the experiments. L.Z., M.C.L., J.R., and A.L. designed the sample. J.R., L.Z., A.D.W., and A.L. grew the sample. C.S., L.Z., G.N.N., and M.C.L. fabricated the sample. L.Z., M.C.L., C.S., G.N.N., A.J., and R.J.W. analysed the data. M.C.L., L.Z., and R.J.W. wrote the manuscript with input from all the authors.

## Competing interests

The authors declare no competing interests.
