## [Peer Review File · Nature Communications]

REVIEWER COMMENTS

Reviewer #1 (Remarks to the Author):

The authors report ultra-low noise GaAs/AlGaAs QDs. The results are impressive showing a close to lifetime limited QD emission linewidth and long electro-spin lifetime. The QDs emit in a wavelength range relevant for short-range networks and quantum memories.

The authors made this fundamental technological advancement in QD photonics by making use of:
1) innovative self-assembly procedure for the QDs, which are realized by local droplet etching, thus making possible the matching the emission with Rb-based photon memories and strongly reducing strain in the QDs;

2) carefully designing the GaAs/AlGaAs p-i-n structure in terms of doping and Al concentration, this way strongly reducing charge noise.

The paper is clear, the experimental data detailed and sound and the results groundbreaking.

I recommend a couple of additions:

1) the authors claim that the results are not coming from a single "hero" QD, showing a second QD which demonstrates similar results. This is remarkable but I would like to see something more quantitative, like an estimation of probability to find a QD with the good properties in the ensemble, based on the group experience with the sample.

2) The careful design of the Al content to make DX centers above the conduction band edges in the large part of the device is a smart way to reduce charge noise. However, it is not sufficient. To be effective, this should be accompanied by a negligible impurity related trap density in the barrier. To make the results reproducible, the assessment of the impurity related trap density incorporated unintentionally in the epilayer should be reported in the Supplementary Materials.

Reviewer #2 (Remarks to the Author):

The manuscript entitled „Low-Noise GaAs Quantum Dots for Quantum Photonics“ by L. Zhai reports on the realization and optical study of high-quality GaAs quantum dots (QDs). These type of self-assembled QD have turned out to be of high interest for the generation of entangled photon pairs at wavelengths below 800 nm and they are interesting candidates to be interfaced with Rb atoms in hybrid quantum systems, like quantum memories. Despite these appealing properties of GaAs QDs (realized by local droplet etching) they have suffered so far from blinking and rather broad emission linewidths, well above the homogeneous linewidth, as the authors point out in the introduction of their manuscript. To tackle this issue the authors developed a layer design of AlGaAs heterostructures with low Al content in the vicinity of the QDs to suppress detrimental effects of DX states, leading e.g. to blinking. Moreover, they apply electrical fields in growth direction to control and stabilize the charge states of the studied QDs.

The manuscript is of very high technological and experimental quality and it addresses a timely important and interesting topic which will certainly be interesting for the readership of Nature Communications. Moreover, the manuscript is well written, and the conclusions are sound. Therefore, I can, in principle, recommend publication of the work in Nature Communications.

Before possible publication in Nature Communication the authors should address the following points in a revised version of their manuscript:

1. Introduction: "telecoms wavelengths" should read "telecom wavelengths", instead of "have high

efficiency (800 nm)" it's maybe better to mention the wavelength (range) where SI-APDs have highest efficiency, "to extend the wavelength range" could maybe be extended to "to extend the wavelength range towards both, short and long wavelengths" (or something like this).

2. Introduction: Some more details, or at least proper references could be added on "facilitating short-range networks. I.e. which distance is meant by "short-range" and would it be free-space communication?

3. Introduction: In the section "On short time scales ... photon stream" the authors could mention the impact of the excitation scheme, i.e. non-resonant, wetting-layer resonant, p-shell resonant, s-resonant, on the charge noise.

4. Fig. 1 and discussion: The authors should also mention (typical emission linewidths under non-resonant excitation used on panel (b) (to compare them with values under resonant excitation). In panels (a) and (b) the electrical field should be added as additional x-axis. The about three orders of magnitude higher count rates in resonance fluorescence should be mentioned and explained.

5. QD properties: Possibly the main advantage of droplet etched QDs is their low fine-structure splitting (FSS). In this regard the value mentioned in the SI should also be included in the main text. Moreover, I think it would be very important to include a FSS statistics of the QDs in the present sample to get a better idea whether ultra-small charge noise and low FSS can be achieved simultaneously.

6. Sample description and table 1: the doping concentration (of the different layers) should be mentioned. What is the QD density of the present sample?

7. Page 2: The authors mention "all Al-concentrations in this design are small enough that processing into micropillars ...". In this regard I wonder if the electrical design including the necessary layer thicknesses would be compatible with the design of electrically contacted QD micropillars reported e.g. in C. Böckler et al. Appl. Phys. Lett. 92, 091107 (2008). Please comment.

8. waveguides will not be hindered by oxidation

9. Page 3: Regarding "Such a wide range of charge tuning... wavelengths" it would be good to explain the reason for the wide range of charge tuning in the present sample.

10. Regarding the E-field fit: I assume the permanent dipole is oriented along the growth direction. This should be mentioned e.g. via "the permanent dipole moment in growth direction". What is the build-in voltage of the device? This information should be included in the discussion.

11. Fig. 2 and discussion: In addition to the decay rate also the spontaneous lifetime should be mentioned (explicitly). Panel (d) it may be interesting to calculate the ratio of "measured linewidth/lifetime limit" and plot it as additional y-axis in panel (d). Panel (e) it seems the g^2 -background is suppressed by the chosen scaling. Is it really zero between the peaks? Panel (g) I recommend using ms instead of s in the x-scaling. I think the indicated zoom-in to panel (f) is not ideal because at first sight it suggests a zoom in into the time range around 5 ns.

12. Page 4: Regarding the remaining noise and spectral broadening I wonder if noise from the voltage source could explain (part) of the remaining noise. Please comment.

13. Page 4: An error should be given for the g^2 -value 2%. How was this value obtained?

14. Page 5: Also for the g-factors, for the spin-relaxation time and for the initialization fidelity error need to be given.

15. Methods: The description of the Methods is rather short and could be possibly be extended, especially regarding the spectroscopic setup.

Reviewer #3 (Remarks to the Author):

In this work, the authors embed GaAs droplet quantum dots within a n-i-p diode in order to show several basic building blocks for future device development based on such structures. They first show the basic ability to access a desired charge state by adjusting the applied gate voltage. They then perform a series of measurement under resonant excitation of the negatively charged trion state: resonant excitation to determine the transition linewidth (in a time-averaged sense), time-resolved PL to determine the excited state lifetime, linewidth of the emission under resonant excitation, and photon antibunching, both to look at single-photon purity (statistics at zero time delay) and blinking (statistics at long time delay). Finally, they perform measurements looking at spin initialization and spin lifetime.

This work is clearly of high-quality, in terms of the scope of the different measurements performed and the results obtained. GaAs droplet quantum dots represent one of the most promising developments from the materials side, as far as single quantum emitter systems are concerned, and this work goes some way in showing that the performance can come close to that of the more widely studied In(Ga)As Stranski-Krastanov (SK) quantum dots.

Overall, I feel that the work is well-suited for publication in Nature Communications, though I have a few comments I would like the authors to address below:

1) As noted in the introduction, the fine structure splitting (FSS) for GaAs droplet QDs is typically significantly smaller than in In(Ga)As SK quantum dots, to the extent that in Ref. 3, no strain tuning to compensate for the FSS was needed. Can the authors comment on their relatively large FSS values for droplet QDs?

2) In Fig. 2e, the authors should state their uncertainty on the $g_2(0)$ value, what it represents (e.g., one standard deviation), and how $g_2(0)$ is normalized in their data. How does the cw $g_2(\tau)$ change at long time delays change if they pump closer to saturation? Presumably this may be more relevant if the QD is to be operated as a (high rate) on-demand single photon source.

3) Can the authors comment on the ability to retain the GaAs droplet QD optical performance in the typical thinner device layers associated with integrated photonics devices like photonic crystals, etc? Would a different diode structure be required?

4) In other works, both the authors and other groups have been able to measure the spectrum of QD frequency fluctuations (e.g., through a heterodyne measurement). Can the authors comment on applying such a technique to getting a better idea of the time scales of the fluctuations that cause broadening beyond the radiative limit?

5) I disagree with the authors comment that their inference (from g_2 measurements) that blinking is absent is 'a significant improvement to all previous GaAs quantum dots'. In particular, reference 3 also

used GaAs quantum dots, and did not observe blinking, while also observing significantly higher brightness levels than in the current work (unsurprising due to the cavity enhancement). That work also demonstrated fairly high single-photon indistinguishability and essentially pure single photons under π pulse excitation. Of course, their indistinguishability is helped by Purcell enhancement. Nevertheless, I think the authors should more properly cite this previous work, and be more careful regarding claims of blinking, etc.

We are much encouraged that our work is appreciated by all three referees:

Referee 1 -- “The results are impressive showing a close to lifetime limited QD emission linewidth and long electro-spin lifetime. [...] The paper is clear, the experimental data detailed and sound and the results groundbreaking.”

Referee 2 -- “The manuscript is of very high technological and experimental quality and it addresses a timely important and interesting topic which will certainly be interesting for the readership of Nature Communications.”

Referee 3 -- “This work is clearly of high-quality, in terms of the scope of the different measurements performed and the results obtained. [...] Overall, I feel that the work is well-suited for publication in Nature Communications.”

The referees made a few specific comments. We respond to these points below. We have uploaded a revised version of the manuscript and an additional pdf-file in which all changes are indicated.

Referee 1:

1. the authors claim that the results are not coming from a single “hero” QD, showing a second QD which demonstrates similar results. This is remarkable but I would like to see something more quantitative, like an estimation of probability to find a QD with the good properties in the ensemble, based on the group experience with the sample.

We thank the referee for making this very good point here. To investigate this question, we performed linewidth and lifetime measurements on a few more quantum dots. The results of these measurements are summarised in a new figure (Extended Data Fig. 4). Based on the new data, there is a relatively high chance to find a quantum dot with excellent optical properties emitting below 785 nm, a wavelength range which covers the rubidium D₂ line. Out of ten random quantum dots in this wavelength range, more than seven have close-to transform-limited optical linewidths. However, for

quantum dots emitting above 785 nm the chance of close-to transform-limited optical linewidths becomes smaller. We think this is related to the 10 nm thick $\text{Al}_{0.33}\text{Ga}_{0.67}\text{As}$ -layer into which the quantum dots are grown by local droplet etching. The longer the emission wavelength, the deeper the quantum dot tends to be. When a quantum dot depth comes close to 10 nm, the optical properties could be affected by the $\text{Al}_{0.33}\text{Ga}_{0.67}\text{As}/\text{Al}_{0.15}\text{Ga}_{0.85}\text{As}$ interface. We point out this issue in the Methods section, and suggest a simple way to avoid it by making the $\text{Al}_{0.33}\text{Ga}_{0.67}\text{As}$ -layer below the quantum dots 5 nm thicker.

2. The careful design of the Al content to make DX centers above the conduction band edges in the large part of the device is a smart way to reduce charge noise. However, it is not sufficient. To be effective, this should be accompanied by a negligible impurity related trap density in the barrier. To make the results reproducible, the assessment of the impurity related trap density incorporated unintentionally in the epilayer should be reported in the Supplementary Materials.

We appreciate the referee's comment. High-quality material is indeed an important element for achieving low-noise quantum dots. In the molecular beam epitaxy chamber used here, we estimate a background impurity concentration of $\sim 5 \times 10^{14} \text{ cm}^{-3}$ ($\pm 2 \times 10^{14} \text{ cm}^{-3}$) for the $\text{Al}_{0.15}\text{Ga}_{0.85}\text{As}$ -layers. We have included this information in the Methods section (Table 1). This estimation is made based on the measurements in Ref. [50], where the background impurity concentrations accounting for all charged impurity scattering centres in GaAs and $\text{Al}_{0.33}\text{Ga}_{0.67}\text{As}$ layers are determined: $N_{\text{Al}_{0.33}\text{Ga}_{0.67}\text{As}} = 7.8(3) \times 10^{14} \text{ cm}^{-3}$, $N_{\text{GaAs}} = 2.6(1) \times 10^{14} \text{ cm}^{-3}$.

Referee 2:

1. Introduction: "telecoms wavelengths" should read "telecom wavelengths", instead of "have high efficiency (800 nm)" it's maybe better to mention the wavelength (range) where SI-APDs have highest efficiency, "to extend the wavelength range" could maybe be extended to "to extend the wavelength range towards both, short and long wavelengths" (or something like this).

We thank the referee for these comments. We have changed "telecoms wavelengths" to "telecom wavelengths". Regarding the high detection efficiency, we refer to a wavelength range of 600 nm – 800 nm in the revised manuscript. We have also modified the phrase "to extend the wavelength range" as the referee suggested.

2. Introduction: Some more details, or at least proper references could be added on "facilitating short-range networks. i.e. which distance is meant by "short-range" and would it be free-space communication?

We agree that our statement about short-range networks is a bit vague. We thank the referee for pointing it out. We decided to remove the phrase "facilitating short-range networks" in the new version of the manuscript.

3. Introduction: In the section "On short time scales ... photon stream" the authors could mention the impact of the excitation scheme, i.e. non-resonant, wetting-layer resonant, p-shell resonant, s-resonant, on the charge noise.

We thank the referee for mentioning the impact of the excitation scheme. In literature, it has been shown that a weak non-resonant laser plays a role in stabilising the charge environment of the

quantum dot (Ref. [22]). We have added a corresponding sentence in the revised manuscript. It is worth mentioning that for GaAs quantum dots in the bulk, such as the ones in Ref. [22], an additional weaknon-resonant laser was essential for resonance-fluorescence-type measurements – it provides some control of the charge fluctuations. For the GaAs quantum dots in our sample, a non-resonant illumination is not required.

Motivated by the referee's comment and another question from referee 3, we have produced an additional figure (Extended Data Fig. 5) presenting $g^{(2)}$ -measurements with different excitation schemes. Under above-band and p-shell excitations, the $g^{(2)}$ -measurement (left panels in Extended Data Fig. 5) shows no telegraph noise even out to millisecond timescale, similar to the resonant case. This rules out any blinking behaviour on time-scales less than tens of milliseconds, confirming that the quantum dot is a stable quantum emitter. Moreover, we also performed $g^{(2)}$ -measurements using resonant π -pulse excitation. This excitation scheme is the most relevant one for many applications including single-photon sources. The pulsed $g^{(2)}$ is carefully normalised by a mathematically justified method which is detailed in both the figure caption and the Methods section. Importantly, the heights of side-peaks in the pulsed $g^{(2)}$ -measurement all stay very close to one, showing that for the π -pulse excitation scheme there is also an absence of the blinking.

4. Fig. 1 and discussion: The authors should also mention (typical emission linewidths under non-resonant excitation used on panel (b) (to compare them with values under resonant excitation). In panels (a) and (b) the electrical field should be added as additional x-axis. The about three orders of magnitude higher count rates in resonance fluorescence should be mentioned and explained.

We thank the referee for these comments. We have added the electric field as an additional x-axis in panel (b) in Fig. 1. We have also added an upper bound on the linewidth under non-resonant excitation to the caption of Supplementary Fig. S1. This upper bound (7.7 GHz) is estimated by comparing the spectrum width of the X^{1-} emission and a laser. We found that in the end we are limited by the resolution of our spectrometer.

The three orders of magnitude discrepancy in count rates between resonance fluorescence and photoluminescence in Fig. 1 is mostly due to the difference in excitation powers. The excitation power used for resonance fluorescence corresponds to a Rabi frequency $\Omega \sim 0.5 \Gamma$ — in saturation, we have raw count rates up to 6.5 MHz (new figure: Supplementary Fig. S1). For photoluminescence, we typically use excitation powers far below saturation. Under weak non-resonant excitation power, we can determine the relevant positions of each charge state in gate voltage. When increasing the non-resonant laser power, other charge states are excited and the charge plateaus shift due to the creation of space charge in the device. Besides, the count rates in Fig. 1 panel (b) are recorded by a spectrometer, the count rates in panel (c) are measured by a superconducting single-photon detector with higher detection efficiency. This adds to the discrepancy between resonance fluorescence and photoluminescence count rates.

To show that higher count rates can be reached in photoluminescence, we performed a power dependence measurement on QD2, where we pass the photoluminescence through a grating-based filter and to a superconducting single-photon detector (Fig. S1(d)). We have added a power dependence of the resonance fluorescence and photoluminescence to the Supplementary Information (Fig. S1). We have also added some additional information to the caption of Fig. 1.

5. QD properties: Possibly the main advantage of droplet etched QDs is their low fine-structure

splitting (FSS). In this regard the value mentioned in the SI should also be included in the main text. Moreover, I think it would be very important to include a FSS statistics of the QDs in the present sample to get a better idea whether ultra-small charge noise and low FSS can be achieved simultaneously.

We agree with the referee that a small fine-structure splitting is an important advantage of droplet-etched quantum dots, particularly for the generation of polarisation-entangled photon pairs. We have included the fine-structure splittings of several additional quantum dots in Extended Data Fig. 4(c), together with their linewidth information. We find that in our sample, there is a high chance of finding a QD (< 785nm) with good optical properties (e.g. transform-limited linewidth) in combination with a low fine-structure splitting (< 2 GHz). These results are discussed in the Methods section in the revised manuscript. We also mention the range of fine-structure splittings and linewidths in the main text.

6. Sample description and table 1: the doping concentration (of the different layers) should be mentioned. What is the QD density of the present sample?

The doping concentration of the n+ back-contact is $2 \times 10^{18} \text{ cm}^{-3}$, while for p+ and p++ layers it is around $2 \times 10^{18} \text{ cm}^{-3}$ and $8 \times 10^{18} \text{ cm}^{-3}$, respectively. The QD density of the sample is estimated through spatially resolved photoluminescence measurement in a $25 \times 25 \mu\text{m}^2$ region. By dividing the number of QDs inside this region by the area, we obtain a QD density of $n_{\text{QD}} = 0.37 \pm 0.01 \mu\text{m}^{-2}$. We have added this number to the main text.

7. Page 2: The authors mention “all Al-concentrations in this design are small enough that processing into micropillars ...”. In this regard I wonder if the electrical design including the necessary layer thicknesses would be compatible with the design of electrically contacted QD micropillars reported e.g. in C. Böckler et al. Appl. Phys. Lett. 92, 091107 (2008). Please comment.

We thank the referee for bringing this reference to our attention. We think that a device as mentioned in this reference could be fabricated with a heterostructure similar to ours. To achieve tunnel-coupling between the quantum dots and a back gate, it would be advantageous that the bottom mirror is n-doped such that no contacting from the side of the pillar is required. We have included the reference in the new version of our manuscript.

8. waveguides will not be hindered by oxidation

We made that statement more general by replacing “waveguides” by “nanostructures” in the new version of the manuscript. The material with the highest Al-concentration that we use is $\text{Al}_{0.33}\text{Ga}_{0.67}\text{As}$. It has been previously used as barriers for InGaAs-based waveguides and oxidation was not a limiting factor. We have added a corresponding citation in the new version of the paper (Phys. Rev. B 96, 165306 (2017)).

9. Page 3: Regarding “Such a wide range of charge tuning... wavelengths” it would be good to explain the reason for the wide range of charge tuning in the present sample.

This is an interesting question. We think that the large range of charge tuning arises mainly because of the fact that GaAs quantum dots grown by local droplet etching usually have a larger size in comparison to the InGaAs quantum dots grown in the Stranski-Krastanov mode. We have added a

citation to Ref. [8] showing the size of droplet-etched nano-holes as well as an additional citation (Ref. [41]: Phys. Rev. B 88, 045316 (2013)) measuring effective length-scales of InGaAs quantum dots.

10. Regarding the E-field fit: I assume the permanent dipole is oriented along the growth direction. This should be mentioned e.g. via “the permanent dipole moment in growth direction”. What is the built-in voltage of the device? This information should be included in the discussion.

We thank the referee for these two points. We have modified the phrase as “the permanent dipole moment in the growth direction” in the revised manuscript. The built-in voltage in our device is 1.82 V. We have included this information in the caption of Table 1.

11. Fig. 2 and discussion: In addition to the decay rate also the spontaneous lifetime should be mentioned (explicitly). Panel (d) it may be interesting to calculate the ratio of “measured linewidth/lifetime limit” and plot it as additional y-axis in panel (d). Panel (e) it seems the g^2 -background is suppressed by the chosen scaling. Is it really zero between the peaks? Panel (g) I recommend using ms instead of s in the x-scaling. I think the indicated zoom-in to panel (f) is not ideal because at first sight it suggests a zoom in into the time range around 5 ns.

We agree with referee 2 that also a lifetime should be stated. In the caption of Fig. 2, we have added the lifetime explicitly next to the decay rate.

We like the idea of computing the ratio between measured linewidth and the lifetime limit. However, we think that a second y-axis might make Fig. 2(d) a little too crowded. Instead, we decided to mention the ratio in the main text. We also state more clearly what we define as the plateau centre for this estimation. In a new figure (Extended Data Fig. 4), we show the ratio between the measured linewidth and the lifetime limit for in total ten randomly chosen quantum dots.

In the pulsed $g^{(2)}$ -measurement, the coincidence counts do not drop completely to zero between the peaks. This is due to the dark counts from the two APDs. We explain how the value for $g^{(2)}(0)$ is obtained in Extended Data Fig. 5. We also estimate a “worst-case” value $g^{(2)}(0) = 0.036$ when one does not correct for the detector dark counts at all.

We have also removed the indicated zoom-in to panel 2(f) and used ms-units for the x-axis of panel 2(g).

12. Page 4: Regarding the remaining noise and spectral broadening I wonder if noise from the voltage source could explain (part) of the remaining noise. Please comment.

We thank the referee for raising this point. We double-checked the specifications of our low-noise voltage source as well as the manufacturer’s test results. An average noise level of our voltage source is $3.3 \mu V_{pp}$. This number translates to around 4 MHz spectral broadening (for QD1), corresponding to an additional broadening of 0.7% of the lifetime limit. The power broadening effect has also been studied (depicted in Supplementary Fig. S1(c)). We attribute another ~ 1 MHz ($\sim 0.2\%$ of the lifetime limit) to the small power broadening effect. These small numbers show that the measured linewidth broadening has a different origin in our measurements. We explore the potential origins in Extended Data Fig. 6.

13. Page 4: An error should be given for the g^2 -value 2%. How was this value be obtained?

We agree with the referee that an error should be given for the $g^{(2)}(0)$. We have included this error in the new version of the manuscript. The $g^{(2)}(0)$ value is obtained by integrating the counts in the peak at zero time-delay and comparing it to the corresponding values for peaks at non-zero time-delays. To estimate the error, we split the time-tagged file containing the $g^{(2)}$ -measurement in five time-intervals. We determine the value of $g^{(2)}(0)$ for each interval separately and calculate the unbiased standard deviation of these five intervals as the error of the $g^{(2)}(0)$ value.

14. Page 5: Also for the g -factors, for the spin-relaxation time and for the initialization fidelity error need to be given.

The referee makes a good point here and we appreciate it. We have now added the corresponding errors to the manuscript: $g_e = -0.0761 \pm 0.001$, $g_h = 1.309 \pm 0.001$, spin pumping fidelity $F = 98.3 \pm 0.3 \%$, spin-relaxation time $T_1 = 48 \pm 5 \mu s$.

15. Methods: The description of the Methods is rather short and could be possibly be extended, especially regarding the spectroscopic setup.

We agree that there is only a very brief description of the spectroscopic setup in the methods section. Our microscope has a similar design as in Refs. [22,49]. Therefore, we have added a sentence citing these two references where further aspects of the spectroscopic setup are explained.

Referee 3:

1. As noted in the introduction, the fine structure splitting (FSS) for GaAs droplet QDs is typically significantly smaller than in In(Ga)As SK quantum dots, to the extent that in Ref. 3, no strain tuning to compensate for the FSS was needed. Can the authors comment on their relatively large FSS values for droplet QDs?

We thank the referee for this comment. Referee 2 had a very similar question (point 5). We have added Extended Data Fig. 4(c) where we show the fine-structure splitting for several quantum dots. The lowest fine structure splittings and also the average value is comparable to the numbers presented in Ref. [4]. We have included a sentence in the main text mentioning that smaller fine-structure splittings can be achieved by using (111)-oriented substrates or strain-tuning. In this context we have also cited again Ref. [4]. (We were wondering if the referee confused Ref. [3] and Ref. [4] here since only Ref. [4] uses GaAs quantum dots.)

2. In Fig. 2e, the authors should state their uncertainty on the $g^2(0)$ value, what it represents (e.g., one standard deviation), and how $g^2(0)$ is normalized in their data. How does the cw $g^2(\tau)$ change at long time delays change if they pump closer to saturation? Presumably this may be more relevant if the QD is to be operated as a (high rate) on-demand single photon source.

We fully agree that an error on $g^{(2)}(0)$ should be given. Referee 2 has asked a similar question and we refer here to our answer to point 13 of referee 2.

The $g^{(2)}$ -measurement shown in Fig. 2(e) is performed under resonant π -pulse excitation. This measurement configuration corresponds to a high instantaneous Rabi frequency in comparison to the CW-excitation and is the most relevant excitation scheme for building a single-photon source. To investigate if there is blinking at long time delays, we made an additional evaluation of the same data set which is presented in a new figure (Extended Data Fig. 5(c,d)). We derive a normalisation factor by calculating how many coincidence events are expected per pulse in case of a perfectly stable source (i.e. without blinking), and divide the measured coincidence events by this factor. We obtain a histogram plot in which the bars are very close to the ideal limit of one. This shows that the quantum dot is a very stable single photon emitter under strong resonant π -pulse excitation. We have also performed a CW $g^{(2)}$ -measurement when the QD is pumped at a close-to saturation power (data not shown in the manuscript). There is no blinking and the $g^{(2)}$ looks similarly flat on long time-scales compared to the weak CW resonant excitation.

3. Can the authors comment on the ability to retain the GaAs droplet QD optical performance in the typical thinner device layers associated with integrated photonics devices like photonic crystals, etc? Would a different diode structure be required?

We believe that the achievements presented in our work can be transferred to thinner diode structures. A working n-i-p diode structure locks the QD's charge states and protects it from excess charge noise. Layers thickness would need to be adjusted based on band-structure simulations to make the n-i-p diode suitable for single-mode photonic crystal waveguides, etc. We have added a corresponding sentence citing Refs. [3,38] in the outlook.

4. In other works, both the authors and other groups have been able to measure the spectrum of QD frequency fluctuations (e.g., through a heterodyne measurement). Can the authors comment on applying such a technique to getting a better idea of the time scales of the fluctuations that cause broadening beyond the radiative limit?

We thank referee 3 for this very useful suggestion. To investigate the possible origin of the linewidth broadening beyond the radiative limit, we performed the auto-correlation measurements similar to Fig. 2(f, g) with either an additional small magnetic field or a small detuning between the laser and the quantum dot resonance. As shown in Ref. [42], in the former case, the sensitivity to spin noise is enhanced, while in the latter case, the sensitivity to charge noise is enhanced. The results of these two auto-correlation measurements are depicted in Extended Data Fig. 6. When the charge noise sensitivity is enhanced, we observe noise on a millisecond time-scale as a bunching in the auto-correlation. For enhanced spin-noise sensitivity, the $g^{(2)}$ looks flat and stays close to one. This measurement suggests that in our device, charge noise on millisecond time scales is most likely responsible for the additional line broadening. To investigate this question in more detail, we have planned some future measurements of noise spectra and fast scans over the quantum dot resonance as described in Ref. [31].

5. I disagree with the authors comment that their inference (from g^2 measurements) that blinking is absent is 'a significant improvement to all previous GaAs quantum dots'. In particular, reference 3 also used GaAs quantum dots, and did not observe blinking, while also observing significantly higher brightness levels than in the current work (unsurprising due to the cavity enhancement). That work also demonstrated fairly high single-photon indistinguishability and essentially pure single photons under π pulse excitation. Of course, their indistinguishability is helped by Purcell enhancement.

Nevertheless, I think the authors should more properly cite this previous work, and be more careful regarding claims of blinking, etc.

We would like to apologise for the slightly too strong statement. We have removed the phrase “a significant improvement compared to all previous quantum dots”. Reference 3 presents excellent results which do not suffer from blinking. However, in this paper InGaAs quantum dots were used. To make it clear that reference 3 does not suffer from such issues, we also included this reference in the paragraph explaining corresponding achievements on InGaAs quantum dots (this paragraph starts with: “For InGaAs QDs, embedding the QDs in an n-i-p diode has profound advantages: ...”).

We appreciate the excellent results on GaAs quantum dots in reference 4. However, reference 4 exploits mainly the pulsed excitation scheme, which is, to our knowledge, less sensitive to charge noise than narrow-bandwidth CW excitation. Thus, we think a direct comparison to our sample is not straightforward. We want to emphasise the excellent results in reference 4 by adding it to the following sentence: “On short time-scales, the charge environment is static such that successively emitted photons exhibit a high degree of coherence.”

Finally, to better address the referees’ comments and to improve the readability, we added three figures as Extended Data items and changed the order of the figures in Supplementary Information. We also corrected a few typing errors which we found during the revision of our manuscript. All the changes can be found in the pdf-files ending with _diff.pdf.

REVIEWERS' COMMENTS:

Reviewer #1 (Remarks to the Author):

The authors answers to the reviewers' comments are, in my opinion, sound and give the required deeper understanding of the presented data.

Reviewer #2 (Remarks to the Author):

I thank the authors for their very careful consideration of my comments and suggestions and for changing the manuscript accordingly. I think the presentation of their results improved significantly and I can recommend publication of the manuscript in Nature Communications "as is".

Reviewer #3 (Remarks to the Author):

The authors have done an extensive job in responding to my (and all referee) comments, and in doing so have significantly improved their manuscript. In particular, their measurements of ten different QDs with seven showing close to lifetime-limited linewidths, measurements of the QD fine structure splitting, and measurements of the intensity autocorrelation out to ten millisecond timescales showing that blinking is negligible but that some charge noise is present all present useful pieces of information that further make this paper an impactful reference for the community. Overall, this is one of the stronger quantum dot papers I have reviewed for Nature Communications, and I am happy to endorse its publication.